# Diazocarbonyl and Related Compounds in the Synthesis of Azoles

**DOI:** 10.3390/molecules26092530

**Published:** 2021-04-26

**Authors:** Anton Budeev, Grigory Kantin, Dmitry Dar’in, Mikhail Krasavin

**Affiliations:** Institute of Chemistry, St. Petersburg State University, 198504 Peterhof, Russia; st079949@student.spbu.ru (A.B.); g.kantin@spbu.ru (G.K.)

**Keywords:** diazocarbonyl compounds, azoles, atom economy

## Abstract

Diazocarbonyl compounds have found numerous applications in many areas of chemistry. Among the most developed fields of diazo chemistry is the preparation of azoles from diazo compounds. This approach represents a useful alternative to more conventional methods of the synthesis of azoles. A comprehensive review on the preparation of various azoles (oxazoles, thiazoles, imidazoles, pyrazoles, triazoles, and tetrazoles) from diazocarbonyl and related compounds is presented for the first time along with discussion of advantages and disadvantages of «diazo» approaches to azoles.

## 1. Introduction

Diazo compounds are incredibly useful reagents in synthetic organic chemistry due to their extremely versatile and unique reactivity [1,2,3,4,5]. These compounds are able to undergo transformations with the loss of nitrogen (under catalytic, photochemical, or thermal decomposition) or with its retention as a constituent part of a reaction product. Along with diazoalkanes, EWG-substituted diazo compounds (diazo-carbonyl, -sulfonyl, -phosphonyl compounds, diazonitriles) are extensively used in organic synthesis. The latter stand out due to their higher stability, safety, and convenience in handling.

One of the highly demanded applications of diazocarbonyl compounds and other stabilized analogs is the construction of carbo- and heterocyclic systems. Recently, a range of new methods have been developed for the synthesis of various heterocycles from diazo compounds, some of which are described in reviews [6,7,8,9,10,11,12]. Versatility and efficiency of diazo compounds are most clearly manifested in the synthesis of azoles—five-membered heterocycles containing a nitrogen atom and at least one other non-carbon atom of either nitrogen, oxygen, or sulfur. Diazo reagents allow synthesis of azoles of many types with two, three, and four heteroatoms. These heterocycles have found broad application in many fields such as organic electronics, functional materials, explosives, dyes, fluorophores, and especially medicine. According to the Drug Bank, there are over 300 FDA-approved drugs containing azole moiety.

However, to our knowledge, there are only a few reviews on the synthesis of azoles from diazocarbonyl compounds and these reviews are focused on one specific azole class, namely 1,3-oxazoles [13], pyrazoles [14,15], and 1,2,3-thiadiazoles [16]. This review aims to summarize all recent developments in this area covering all types of azoles available from diazocarbonyl and related compounds (oxazoles, thiazoles, imidazoles, pyrazoles, triazoles, tetrazoles, with the exception of 1,2,3-thiadiazoles), as well as to consider earlier useful methodologies not mentioned in other reviews.

## 2. Azoles with Two Heteroatoms

### 2.1. 1,3-Oxazoles

1,3-Oxazole (in brief, oxazole) is a privileged motif for drug design. Oxazole derivatives have shown a broad range of biological activities such as anti-inflammatory, antibacterial, anticancer, antifungal, antiviral, antihyperglycemic, and so on [17,18]. Numerous oxazole medicinal agents are of natural origin. Oxazole alkaloids, including macrolides, cyclic and acyclic peptides, siderophores, and small molecules are found in marine invertebrates, bacteria, fungi, and higher plants [19]. Therefore, a vast array of methods for synthesis of 1,3-oxazoles have been developed to date [20].

Among these methodologies, the following two strategies are rather commonly employed: (i) synthesis of 1,3-oxazoles by formal [3+2] cycloaddition between metal carbenes derived from diazo carbonyl compounds and nitriles, and (ii) metal carbene insertion into the N–H bond of amides with subsequent dehydrative cyclization. These routes have been comprehensively reviewed elsewhere [13,21]. In this section, we chose to present only fairly recent examples of these strategies as well as a few rare approaches to 1,3-oxazoles.

#### 2.1.1. Synthesis of 1,3-Oxazoles by Reaction of Diazocarbonyl Compounds and Nitriles

Since the pioneering work of Huisgen on the formal [3+2] cycloaddition between nitriles and diazo carbonyls [22], this method has become the most frequently employed diazo strategy for the synthesis of 1,3-oxazoles. The main disadvantage of this approach is the need to use an excess amount of the nitrile (often as a solvent). However, some protocols have been developed that allow reducing the excess of the nitrile down to 5-fold.

In 2016, the Park group developed an efficient method for the synthesis of highly substituted oxazoles **3** by gold-catalyzed cycloaddition reaction of *α*-diazo oxime ethers **1** and nitriles **2** (Scheme 1) [23]. The reaction tolerated a broad range of substituents (including alkyl, aryl, heteroaryl, and vinyl) and afforded good to excellent product yields. The formation of the oxazole ring is believed to proceed via the formation of zwitterionic intermediate **1A**, which subsequently undergoes cyclization to oxazole **3**. Interestingly, the cyclization occurred via the attack of the carbonyl group rather than the attack of the oxime ether group on the nitrilium electrophilic center, presumably due to the *Z*-configuration of the C=N double bond (Scheme 1).

In the synthetic exploration of 2-diazoacetyl-2*H*-azirines **5**, which are available from 5-chloroisoxazoles **4**, Khlebnikov and co-workers demonstrated the utility of **5** for the preparation of oxazoles [24]. Rhodium-catalyzed reaction with nitriles **6** afforded oxazoles **7** with preservation of azirine functionality. On the other hand, NiCl_2_-catalyzed reaction with acetylacetone **8** in acetonitrile provided pyrrolyl-substituted oxazoles **9** (Scheme 2).

An unusual tris(pentafluorophenyl)borane catalyst was employed for the reaction of nitriles **10** and diazo carbonyls **11** by Kumaraswamy and Gangadhar [25]. This metal-free protocol was found compatible with many functional groups including alkyl, aryl, alkyl(aryl)oxy, and allowed using a lower (5-fold) excess of the nitrile partners. A plausible reaction mechanism implies initial activation of diazocarbonyl compound **11** by the catalyst generating alkenediazonium salt **3A,** which is attacked by nitrile **10** with extrusion of N_2_ molecule to give intermediate **3B**. The latter undergoes cyclization to form oxazole product **12** (Scheme 3).

Diazo homophthalimides **13** were the first heterocycles containing an *α*-diazocarbonyl moiety, which were involved in a [3+2] cycloaddition with nitriles **14** to deliver oxazole-fused adducts **15** as reported by the Krasavin group [26]. The reaction displayed a generally high functional group tolerance giving low yields only in case of olefinic, benzylic, and sterically hindered nitriles. Diazo homophthalimides **13** are easily available via diazo transfer to the corresponding homophthalimides (Scheme 4).

Further exploring the synthesis of fused oxazoles from diazo containing heterocycles, the same group reported a [3+2] cycloaddition reaction of diazobarbituric acid derivatives **16** (available from barbituric acids via a novel sulfonyl-azide-free (SAFE) diazo transfer protocol [27]) with nitriles **17** [28]. Diversely substituted oxazolo[5,4-*d*]pyrimidine-5,7(4*H*,6*H*)-diones **18** were obtained in generally moderate to good yields. However, mono-*N*-substituted diazobarbituric acid failed to give the desired product. Non-symmetric diazobarbituric acid gave mixture of regioisomers. Notably, the attempt of performing this reaction with diazo Meldrum’s acid **19** led to the formation of oxazolo[4,5-*e*][1,3]oxazinone **20** via acyl ketene intermediate **5B,** which was involved int [4+2] cycloaddition with another molecule of acetonitrile (Scheme 5).

Alicyclic diazocarbonyl compounds have also been employed in oxazole synthesis. Thus, in 2018, Fan and co-workers described the synthesis of 6,7-dihydrobenzo[*d*]oxazol-4(5*H*)-ones **23** by reacting cyclic diazo diketones **21** with nitriles **22** [29]. The best yields were obtained with non-substituted diazo substrates. Analogously to the example discussed above, diazo Meldrum’s acid did not provide the desired oxazole. From the mechanistic standpoint, the reaction likely proceeds via the attack of the nitrile on the rhodium carbene, generated from diazo species, following steps similar to the above-mentioned examples (Scheme 6).

The reaction between polybrominated *o*-quinone diazides **25** (available via diazotization of aminophenols **24**) and nitriles **26** leading to benzoxazoles **27** was described in 2017 by Vasin and co-workers [30]. The reaction proceeded on conventional heating, albeit in modest yields (Scheme 7).

The nitrile cycloaddition strategy has been used for preparation of a number of oxazole-containing biologically active compounds such as cyclin-dependent kinase 2 inhibitors [31], peroxisome proliferator-activated receptor *δ* agonists [32], and atypical antipsychotics [33].

#### 2.1.2. Synthesis of 1,3-Oxazoles by Reaction of Diazocarbonyl Compounds with Amides

The currently well-known approach to oxazoles **31** from amides **28** and diazo compounds **29** was first described by Moody and co-workers in 1996 (Scheme 8) [34,35]. It was developed as an alternative to the cycloaddition of nitriles, which gave unsatisfactory results when applied to *N*-protected *α*-aminonitriles derived by dehydration of *α*-amino acid amides. The new approach involves Rh carbene *NH*-insertion into the amide molecule followed by cyclodehydration via the so-called Wipf protocol, i.e., using a mixture of Ph_3_P, I_2_ and Et_3_N (Scheme 8).

Notably, several methodologies have been developed that allow the synthesis of oxazoles regioisomeric to the ones obtained by the Moody approach. The key step in these alternative methods is the formation of a carbonyl ylide via amide oxygen atom attack on metal carbene rather than an *NH*-insertion as the first step of the synthetic sequence (*vide infra*). Another attractive feature contrasting these protocols to the Moody oxazole synthesis is that the oxazole ring is constructed in one step.

Cu(OTf)_2_-catalyzed synthesis of 2,4-substituted oxazoles **34** from terminal diazocarbonyl compounds **32** and amides **33** was developed by the Subba Reddy group [36]. This method works well for both aliphatic and aromatic substrates providing good product yields. The reaction was unsuccessful only when both substrates were aliphatic. The following mechanism was invoked. Initially, copper carbene intermediate is formed on interaction of **32** with the Cu-catalyst. Rather unexpectedly, the subsequent attack of the amide on the Cu-carbene occurs through the carbonyl group, giving corresponding ylide in preference to the attack by the nitrogen atom, which would result in an *NH*-insertion product. Subsequent protodemetallation and cyclodehydration gives the observed oxazole **34** (Scheme 9).

In 2018, Chen and co-workers reported the involvement of diazo dicarbonyl compounds **35** in a similar reaction under Rh(I) catalysis, which afforded fully substituted oxazoles **37** [37]. In this case, however, the yields were significantly lower for aliphatic substrates in comparison to the previous example. From the mechanistic perspective, rhodium(I) carbene complex **10A** likely reacted with the carbonyl oxygen atom of amide **36a** to form intermediate **10B**, which underwent metal protonation (with regeneration of the catalyst) to afford *β*-ketoester **10C**. The latter entered the cyclization/dehydration sequence, thus producing trisubstituted oxazole **37a**. Preliminary mechanistic studies revealed that the attack on Rh(I) carbenes was likely the rate-determining step (Scheme 10).

Thus, by using either of these methods of the Moody protocol, both oxazole regioisomers can be accessed starting from the same set of substrates. The reasons for such a switch in regiospecificity are not entirely clear. Presumably, it can be explained by the change in electrophilicity of the intermediate metal carbene either upon changing metal, upon changing ligands, or both [38].

#### 2.1.3. Other Approaches to the Synthesis of 1,3-Oxazoles

The reaction of *α*-ketooximes **38** with terminal *α*-diazo carbonyl compounds **39** afforded polysubstituted oxazoles **40** in moderate to good yields as reported in 2018 by Swamy and co-workers [39]. The reaction tolerates many substituents, including alkyl, (hetero)aryl, as well as ester, ketone, and nitrile functional groups. The following reaction mechanism was proposed by the authors. The reaction is believed to be initiated by the interaction between copper carbene **11A** and ketooxime **38a,** which leads to aziridine **11B**. The latter undergoes an intramolecular rearrangement to form intermediate **11C.** Subsequent dehydration of **11C** leads to oxazole **40a** (Scheme 11).

In 2015, Trujillo and co-workers synthesized a series of novel 4-(benzylthio)oxazoles **44** by reacting acyl isothiocyanates **41** with trimethylsilyl diazomethane (TMSCHN_2_) **42** with subsequent treatment of the oxazole thiones **43** with BnBr-DBU [40]. Isolated in moderate yields, compounds **44** were then converted to oxazole sulfonyl chlorides **45** in treatment with *N*-chlorosuccinimide (NCS). BnS-derivatives **44** were formed along with thiadiazoles **46,** which is explained by two possible reaction pathways. In path A, initially formed intermediate **12A** underwent cyclization via the attack of the carbonyl group on the carbon atom adjacent to N_2_^+^, while in path B, intermediate **12A** undergoes cyclization via the attack of the thiocarbonyl group on the N_2_^+^ moiety (Scheme 12).

#### 2.1.4. Total Synthesis of 1,3-oxazole Containing Natural Products

Both nitrile cycloaddition (Scheme 13, path A) and amide NH-insertion/cyclization (Scheme 13, path B) diazo strategies have found application in total synthesis of oxazole-containing natural products. The former strategy works best with simple alkyl and aryl nitriles **49** or in those cases when 5-unsubstituted oxazole is the target. For fully decorated oxazoles, the latter strategy is often preferred. The choice of a particular synthetic approach can also be dictated by the availability of the requisite nitrile **49** or amide **48**. However, both routes are complementary and can be considered interchangeable as was shown by Moody and co-workers [34]. Hence, this subsection deals with selected works by the Moody group that have not yet been reviewed.

Total synthesis of siphonazole **58**, an unusual metabolite from *Herpetosiphon* species [41], involves the construction of the oxazole intermediate containing a bulky styrene substituent (**55**) via an NH-insertion reaction of TBS-protected 3-hydroxy-4-methoxycinnamamide **52** and methyl 2-diazo-3-oxobutanoate **53** followed by the Wipf cyclodehydration. The other oxazole portion of the siphonazole molecule (**57**) was obtained by a [3+2] cycloaddition of **53** with iodoacetonitrile **56**. From oxazoles **55** and **57**, the natural product was assembled in three steps (Scheme 14).

A more complex target polyazole peptide antibiotic goadsporin **68** was synthesized in 2017 [42]. It consists of four oxazole and two thiazole rings, linked through a number of amino acid residues. 5-Substituted oxazoles **60**, **62**, **64** were synthesized from the corresponding amino acids amides **59**, **61**, **63** using NH-insertion/cyclization strategy, whilst 5-unsubstituted oxazole **67** was synthesized by [3+2] cycloaddition strategy from *L*-serine-derived nitrile **65** and ethyl 2-diazo-3-oxopropanoate **66** (Scheme 15).

Similarly, with a goal of synthesizing cyclic dodecapeptides wewakazole **69** and wewakazole B **70 [43]**, 2,4-disubstituted oxazoles **73** and **76** were accessed via the nitrile cycloaddition strategy while 2,4,5-trisubstituted oxazoles **60, 78**, and **80** were prepared by the NH-insertion/cyclization strategy. Nitriles **72** and **75** were obtained by dehydration of amino acid amides **71** and **74**, respectively (Scheme 16).

### 2.2. 1,2-Oxazoles (Isoxazoles)

1,2-Oxazole ring plays an important role in organic synthesis. This heterocycle serves as a masked equivalent of a 1,3-dicarbonyl compound, which can be useful in total synthesis of natural compounds [44]. Moreover, other heterocycles are available from isoxazoles, such as 1,3-oxazoles, azirines [45], pyridines, and pyrroles [46]. Biological profile of isoxazoles is somewhat similar to 1,3-oxazoles [47]. Classic routes to isoxazoles include condensation of hydroxylamine with 1,3-dicarbonyl compounds and 1,3-dipolar cycloaddition of nitrile oxides to alkynes, although many other methods have been developed [46,48]. As for diazo methodologies for the synthesis of isoxazoles, information remains scarce. Only a few reports can be found in the literature to date.

Duffy and co-workers described the preparation of unusual 4*H*-imidazo[4,5-*c*]isoxazole **84** from diazo compound **83** which, in turn, was obtained via the coupling of acid chloride **81** with ethyl diazoacetate **82** [49]. On heating, **83** underwent the Wolff rearrangement to give ketene **17B,** which cyclized to form zwitterionic intermediate **17C**. The latter eliminated a CO_2_ molecule, thereby producing *ortho*-nitroso carbene intermediate **17D**. The electrocyclization that followed afforded imidazoisoxazole **84** (Scheme 17).

In a 2018 work by the Wan group, diazo esters **85** were used for in situ preparation of nitrile oxides, which underwent highly regioselective [3+2] cycloaddition with enols generated from *β*-ketoesters **86 [50]**. The reaction tolerated a broad range of substituents in the aromatic ring and furnished 3,4,5-substituted isoxazoles **87** in moderate to good yields. Based on preliminary studies, authors proposed the following mechanism for the reaction (shown for (Het)Ar = Ph). Presumably, copper carbene **18A** generated from **85** was attacked by *tert*-butyl nitrite to form intermediate **18B**. The latter decomposed to nitrile oxide **18C,** which, in turn, underwent a [3+2] cycloaddition with magnesium enolate **18D** to produce intermediate **18E**. Isoxazole **87** was likely produced via the dehydration of hydroxyisoxazoline **18E** (Scheme 18).

Alkynes **89** can also be effective replacements for *β*-ketoesters in this reaction as demonstrated by Wang and co-workers [51]. The reaction proceeded in a highly regioselective manner even with internal alkynes (isoxazoles **90a**), albeit in lower yield. The authors proposed the following radical mechanism for the reaction. Nitroso radical generated by homolytic cleavage of *tert*-butyl nitrite along with *tert*-butoxy radical is thought to react with Cu(I) carbene intermediate **19A** to give organocopper intermediate **19B**. Heterolytic bond cleavage of **19B** with elimination of copper(0) followed by deprotonation could give nitrile oxide **19D,** which could then undergo a [3+2] cycloaddition to alkyne **89** to produce desired isoxazole **90**. In the course of the reaction, Cu(0) is oxidized to Cu(I) by the *t*BuO radical, which completes the catalytic cycle. Alternatively, deprotonation of intermediate **19B** could occur with subsequent heterolytic C-Cu bond cleavage to give the nitrile oxide intermediate **19D** with the same subsequent events (Scheme 19).

### 2.3. 1,3-Thiazoles

1,3-Thiazole-based materials are now extensively studied for application in organic electronics due to their suitable electronic, optical, and spatial properties. 1,3-Thiazoles have been employed for the preparation of organic field-effect transistors (OFETs), organic photovoltaic cells (OPVs), and organic light-emitting diodes (OLEDs) [52]. Moreover, the thiazole ring has been utilized in the design of medicinal agents for the treatment of viral and bacterial infections, diabetes, Alzheimer’s disease, neglected protozoan diseases, and more [53]. Standard approaches to the synthesis of 1,3-thiazoles, such as thionation of *α*-acylaminoketones (the Gabriel synthesis) and condensation of *α*-halocarbonyl compounds with thioamides (the Hantzsch synthesis) are frequently employed [54,55].

The diazo approaches to thiazoles discussed in this subsection of the review are somewhat similar to these classic methods: (i) reaction of *α*-diazocarbonyl compounds (acting as the *α*-halocarbonyl compounds equivalent) with thioamides or thioureas; (ii) *α*-carbonyl carbene NH insertion into thioamides or thiourea with subsequent thionation; and (iii) less conventional reactions.

#### 2.3.1. Synthesis of 1,3-Thiazoles from Diazocarbonyl Compounds and Thioamides or Thioureas

In 1995, Kim and co-workers developed a method for the synthesis of 2,4-disubstituted thiazoles **93** by BF_3_∙OEt_2_-promoted reaction of ethyl diazopyruvate **91** with thioamides **92** [56,57]. The best yields were obtained with aromatic thioamides, while for aliphatic and heteroaromatic thioamides, lower product yield were obtained. The following mechanism was proposed as plausible: The initially formed boron enolate **20A** could be attacked by thioamide **92** to produce thiocarbonyl ylide **20B**. The thiazole product (**93**) could arise as the result of the cyclodehydration of the intermediate **20B** (Scheme 20).

Later, the same group demonstrated the applicability of this procedure to bisthioamides, namely benzene-1,3-bis(carbothioamide) **94** [58]. Thus, bisthiazolylbenzene **95** was obtained in 83% yield. In this case, the use of MgSO_4_ or molecular sieves as desiccants was essential to avoid the formation of by-products. Compound **95** was used for preparation of unusual thiazole-containing crown ethers **98** (Scheme 21).

In 1990, Capuano and co-workers observed the formation of 2,4,5-substituted thiazoles **101** upon thermolysis or irradiation of 2-diazo-1,3-diketones **99** in the presence of thioamides **100** [59]. Under thermolysis conditions, thiazoles **101** were formed as a mixture with 4*H*-1,3-oxazin-4-ones **102** (with a predominance of the latter), while on irradiation, thiazoles were the only products. Unfortunately, the yields were only modest in both cases. The reaction presumably proceeded via ketocarbene **22A,** which can react with thiourea to produce intermediate **22B**. Subsequent cyclodehydration can afford thiazole **101**. Alternatively, **22A** can undergo the Wolff rearrangement with the formation of *α*-oxoketene intermediate **22C,** which could be attacked by the *NH* moiety of thioamide **100** to give intermediate **22D**. The latter could cyclize with the loss of H_2_S and afford 4*H*-1,3-oxazin-4-one **102** (Scheme 22).

The Villalgordo group employed Cu(I) bromide for this reaction to selectively prepare thiazoles **105** in good yields [60]. Further extending the scope of this methodology, functionalized 2-aminobenzothioamide **106** was used to obtain thiazoles **107**, albeit in lower yields. The use of aromatic thioamides proved to be crucial for this reaction since aliphatic ones failed to give the desired products. The tentative mechanism involves the reaction of copper carbene **23A** (generated from *α*-diazo-*β*-ketoester **103** on action of CuBr) with the thioamide **104** to give an intermediate **23B** whose cyclodehydration produced thiazole **105** (Scheme 23).

Rhodium catalysis was also successfully employed in the synthesis of thiazoles. In 2010, the Moody group reported the reaction of diversely substituted 1-diazopropan-2-ones **108** with aromatic thioamides **109** catalyzed by rhodium perfluorobutyramide (Rh_2_(pfm)_4_), which afforded fully substituted thiazoles **110** in generally good yields (Scheme 24) [38]. The mechanism is similar to that shown in Scheme 23. Other Rh(I) catalysts, namely by [Rh(COD)Cl]_2_, were also found suitable for promoting this reaction [37].

Trifluoromethyl-substituted thiazoles **113** and **115** were synthesized by Obijalska and co-workers via the BF_3_·OEt_2_-promoted condensation of 3-diazo-1,1,1-trifluoropropan-2-one **111** and thiourea **112** or benzothioamides **114** (Scheme 25) [61]. The reaction proceeded smoothly with thiourea, but when thioamides were used, dehydration under MsCl/Et_3_N conditions was additionally needed. To overcome this drawback, heating at 150 °C under the microwave irradiation was alternatively performed, which also gave slightly higher yields of the desired 2-arylthiazoles **115**. As in some of the previous examples, the aliphatic thioamides failed to give any thiazole products. 2-Aminothiazole **113** was also furtherly derivatized to obtain CF_3_-analogs of immunomodulatory drug candidate Fanetizole (**117**) and anti-inflammatory drug Lotifazole (**119**) (Scheme 25).

The Narsaiah group reported another example of the coupling of diazo compounds **120** with thiourea **112** under Cu(OTf)_2_ catalysis [62]. This protocol worked well for a variety of aliphatic, donor- and acceptor-substituted aromatic, and benzylic diazoketones, affording 2-aminothiazoles **121** in excellent yields. A catalyst-free version of this reaction using PEG-400 as a solvent was also reported (Scheme 26) [63].

#### 2.3.2. Synthesis of 1,3-Thiazoles Through the Reaction of Diazocarbonyl Compounds and Amides followed by Cyclization

In 2004, the Moody group demonstrated the generality of their approach towards 1,3-oxazoles via NH-insertion/cyclodehydration cascade (Section 2.1.2, Scheme 8) for synthesis of other 1,3-azoles, such as thiazole and imidazole (Section 2.4.1,
Scheme 35) [64]. Thus, fully substituted thiazoles **125** were obtained in varying yields via the treatment of *α*-(acylamino)ketones **124** with Lawesson’s reagent in refluxing THF. The approach demonstrated high functional group tolerance in the amide moiety, also working well even with aliphatic amides (Scheme 27).

This approach has found application in the total synthesis of natural products. One of the six thiazoles (**127**) required for the construction of thiopeptide antibiotic amythiamicin D (**128**) was prepared using this methodology from aspartic acid amide derivative **126** and methyl 2-diazo-3-oxobutanoate **53** (Scheme 28) [65,66]. This route was also involved in total synthesis of other thiazole-containing antibiotics, such as GE2270A and GE2270T [67].

#### 2.3.3. Other Approaches to 1,3-Thiazoles Starting from Diazo Compounds

An interesting procedure for the chemoselective synthesis of 2,4,5-substituted thiazoles from diazo carbonyls **129** and *α*-(*N*-hydroxy/aryl)imino-*β*-oxodithioesters **130**/**132** was reported in 2017 by Srivastava and co-workers [68]. The [Cu(CH_3_CN)_4_]PF_6_-catalyzed reaction gives access to both 5-mercaptothiazoles **131** and 5-mercapto-2,3-dihydrothiazoles **133** in good to excellent yields. Mechanistically, the initially formed Cu-carbene **29A** was chemoselectively attacked by thiocarbonyl sulfur of *α*-imino-*β*-oxodithioester **130**/**132** to give thiocarbonyl ylide **29C**, which, in turn, reacted with diazocarbonyl compound **129** to afford intermediate **29D,** thereby regenerating copper carbene species **29A**. Intermediate **29D** underwent intramolecular cyclization producing 2,3-dihydrothiazole **133**. If *N*-hydroxy-substrate **130** was used, **133** was converted to aromatic thiazole **131** with a loss of water molecule on moderate heating (Scheme 29).

### 2.4. Imidazoles

Imidazole derivatives are widely used in medicinal chemistry. Agents based on imidazole moiety possess anticancer, antifungal, antibacterial, antiviral, anti-inflammatory, and other biological activities [69]. Another area of application of imidazoles is in organic light-emitting diodes (OLEDs) and semiconductors [70]. To date, many methods have been developed for the synthesis of imidazoles, including the classical Debus-Radziszewski imidazole synthesis and reaction of *α*-halo carbonyl compounds with amidines [71,72]. However, syntheses based on diazocarbonyl compounds have not yet become widespread.

There are two main approaches to the preparation of imidazoles using diazocarbonyl compounds: Metal catalyzed insertion into the NH bond of ureas followed by cyclization with formation of 2-imidazolones (which can be easily converted to substituted imidazoles), and NH-insertion into amides followed by cyclization in the presence of amine or an ammonia source. In addition, there are also some more exotic syntheses making use of isocyanides, imines, and nitriles.

#### 2.4.1. Synthesis of Imidazoles by Reaction of Diazocarbonyl Compounds with Amides and Ureas

In 2003, the Janda group reported the one-pot procedure for the synthesis of 2-imidazolones **137** by rhodium(II)-catalyzed NH-insertion involving *α*-diazo-*β*-ketoesters **134** and primary ureas **135** followed by TFA-mediated cyclodehydration of insertion products **136** (Scheme 30) [73]. The reaction resulted in good yields with a full chemoselectivity.

Encouraged by the successful results obtained by performing these syntheses in solution, the authors investigated this strategy on solid phase using hydroxypentyl-JandaJel polymer-bound *α*-diazo-*β*-ketoesters **138** (Scheme 31). Imidazolones **140** were then cleaved from the resin either by transesterification to give esters **141** or by the amidation reaction to give more diversely substituted amides **142** in fair yields and high purity. The only limitation was the unallowable use of diazo substrates bearing long alkyl R^1^ substituents. In these cases, the yield was poor, or the product could not be isolated.

In a continuation of this topic, a work on the expansion of the range of suitable diazo carbonyls and further functionalization of imidazolones by the same authors was published in 2004 (Scheme 32) [74]. *α*-Diazo-*β*-ketophosphonates, *α*-diazo-1,3-diketones, *α*-diazo-*β*-ketoamides, diazomalonic ester, ethyl diazoacetate, amd ethyl 2-diazo-2-(diethoxyphosphoryl)acetate have been successfully involved in the reaction. It should be noted that some of the intermediates **144** could be isolated. The resulting imidazolones **145** could be alkylated both in solution (Scheme 33a) and on solid phase (Scheme 33b). In the second case, both transesterification and amidation can be used to cleave the product from the polymer resin.

Another aspect of this work was the conversion of imidazolones **145** to 2-bromoimidazoles **148** by the treatment with POBr_3_ in refluxing benzene or toluene. Compounds **148** can be further derivatized via the Suzuki cross-coupling reaction (Scheme 33c).

Thus, in this series of works, the authors have shown the applicability of this approach for the synthesis of libraries of various imidazolone and imidazole derivatives, also in an array mode.

The above method has found application in medicinal chemistry [75,76]. In 2015, Gong and co-workers obtained a number of imidazolones **152,** which were then used in the synthesis of a series of potential c-Met kinase inhibitors **153** (Scheme 34) [75].

In 2004, the aforementioned Moody’s strategy for the synthesis of 1,3-oxazoles based on the NH-insertion of rhodium carbene intermediates generated from *α*-diazocarbonyl compounds into primary amides (Section 2.1.2, Scheme 8) was extended to the synthesis of 1,3-thiazoles (Section 2.3.2, Scheme 27) and imidazoles [64]. Thus, imidazoles **156** were obtained by treatment of intermediate **155** with a primary amine or ammonium acetate in the presence of acetic acid under reflux conditions (Scheme 35).

#### 2.4.2. Other Syntheses of Imidazoles from Diazo Compounds

In 2017, Zhao and co-workers devised an effective copper-catalyzed cascade cyclization reaction of isocyanides **157** with *α*-diazocarbonyls **158** [77]. This protocol furnishes imidazolines **159** (Scheme 36, a) and biimidazoles **160** (Scheme 36b) in good yields under very mild conditions. Imidazolines **159** were obtained as a mixture of diastereomers using ethyl isocyanoacetate **157a**. Stereochemistry of the product depends on the R^2^ substituent: Bulky hydroxyalkyl R^2^ substituents gave the mixture with predominance of the *trans*-isomer whereas aryl substituents afforded the product mixture with predominance of the *cis*-isomer. Only in the case of the ethoxy group was the *trans*-isomer obtained exclusively. Biimidazoles **160** were obtained when tosyl methyl isocyanide (TosMIC) **157b** was used as the isocyanide component. 

A plausible reaction mechanism is presented in Scheme 37. Initially, the copper isocyanide complex **161** is formed in which the acidity of the *α*-H atom is significantly enhanced compared to the non-complexed **157**. Subsequently, nucleophilic attack of **161** onto the terminal nitrogen atom of *α*-diazocarbonyl compound **158** occurs under basic conditions to give intermediate **37A**. Intermediate **37B**, (generated by the formal [3+2] cycloaddition reaction of **37A** with complex **161**) then undergoes intramolecular cyclization to give the bicyclic intermediate **37C**. Imidazoline **159** is formed from intermediate **37C** via ring-opening followed by HCN elimination and protonation. When TosMIC was used, imidazoline **159** suffered base-mediated elimination of sulfinic acid to produce the intermediate **37E,** which then underwent a formal [3+2] cycloaddition with the third equivalent of complex **161**. The following elimination of sulfinic acid from the intermediate **37F** gave the biimidazole **160**.

In 2016, the reaction of *α*-diazo oxime ethers **162** (available from *β*-ketoester via one-pot oxime formation with subsequent diazo transfer reaction [78]) with nitriles **163** in presence of copper triflate (Cu(OTf)_2_) catalyst was described by Kuruba and co-workers [79]. The developed procedure provided fully decorated *N*-methoxyimidazoles **164** and worked well with a broad range of nitriles, both aliphatic and aromatic. Mechanistically, two possible reaction pathways were proposed. In both cases, the reaction begins with the formation of metal carbene **38A** from diazo compound **162**. In Path 1, carbene **38A** is attacked by nitrile **163** to form adduct **38B,** which affords ylide **38C** after the release of the catalyst. Ylide **38C** then undergoes cyclization giving imidazole **164**. Path 2 leads to the formation of azirine **38D** with subsequent ring expansion. However, no azirine intermediates have been detected or isolated, so it was concluded that path 1 was favored over path 2 (Scheme 38).

Imidazolium salts are currently being intensively studied as electrolytes, metal ligands, ionic liquids, and liquid crystals [80,81,82]. However, only a few studies have been devoted to the synthesis of polyarylated imidazolium salts. Thus, development of new methods for the synthesis of polyarylated imidazolium salts would be a timely undertaking. To achieve this goal, the Liu group developed a gold-catalyzed [2+2+1] annulation reaction involving two equivalents of imine **165** and one equivalent of aryl diazoacetonitrile **166 [83]**. Introducing electron-rich aryl substituents in the imine component resulted in higher yields of imidazolium salts **167** compared to the imines bearing electron-deficient substituents. The same effects were observed from the substituents in the diazo nitrile aryl ring. Moreover, these imidazolium salts can be further modified by the Sonogashira reaction to afford polyalkynyl-substituted derivatives **168** (Scheme 39).

Based on DFT-calculations, the following mechanism was proposed. Imine ***cis*-165** attacks gold carbene **40A** to produce iminium intermediate **40B** which is, in turn, attacked by another imine molecule to give intermediate **40C** (or its conformer **40C’**). The latter can evolve along two possible pathways. Path **a** involves the formation of imidazolidine **40D**. However, this path is unlikely to occur according to DFT-calculations. In path **b**, **40C’** loses HCN to form a stable gold carbene **40E,** which then undergoes 6π-electrocyclization in disrotatory mode. **40F** is affected by a neighboring cation to release LAu^+^. Zwitterion **40G** is subsequently oxidized to desired compound **167**. It is worth mentioning that in this case, aryl diazoacetonitriles can serve as arylmethine fragment source (Scheme 40).

#### 2.4.3. Fused Imidazoles from Diazo Compounds

The most common fused imidazole derivative being synthesized from diazo compounds is imidazo[1,2-*a*]pyridine. The classic approach to this heterocyclic ring is the condensation of 2-aminopyridines and *α*-haloketones or their synthetic equivalents [84]. The imidazo[1,2-*a*]pyridine-based drugs have found application in therapy of musculoskeletal, gastrointestinal, CNS disorders, cardiovascular, and infectious diseases [85]. Little information is available on the synthesis of other fused imidazole systems from diazo compounds.

The first example of coupling of *α*-diazoketones **169** with 2-aminopyridines **170** was reported in 2007 by Yadav and co-workers [86]. The Cu(OTf)_2_-catalyzed reaction proceeded regioselectively and afforded imidazo[1,2-*a*]pyridines **171** in high yields. Both aromatic and aliphatic diazo ketones ere suitable partners for this reaction (Scheme 41). Later, in 2014, this method was employed towards the synthesis of glucagon-like peptide-1 receptor agonists [87].

In 2013, the Gevorgyan group in their work on three-component coupling reaction of 2-aminopyridines **172**, aldehydes **173**, and diazo compounds **174**, demonstrated that the product of this reaction **175** can be transformed to imidazo[1,2-*a*]pyridine **176** [88]. The reaction proceeds via intermediate **177** (formed via a AgBF_4_-catalyzed 1,2-hydride shift) followed by I^+^-mediated cyclization, tautomerization, and elimination of HI (Scheme 42).

An interesting example of using a banana ash extract and dimethylsulfoxide (WEB-DMSO) as solvent cum reagent system for three-step one-pot synthesis of imidazo[1,2-*a*]pyridines **180** was reported by Dutta and co-authors in 2019 [89]. Initially, a WEB-DMSO-promoted aldol condensation of ethyl diazoacetate **82** and aldehyde **178** occurred to give diazo alcohol **181,** which was converted to *β*-ketoester **182** by a Pd-catalyzed 1,2-H shift. Treatment of **182** with NBS and aminopyridine **179** on the third step afforded the corresponding imidazo[1,2-*a*]pyridine **180** (Scheme 43).

A work by the Lee group shows that imidazopyridines can be synthesized not only from aminopyridines, but also directly from pyridines **183** by a formal aza-[3+2] cycloaddition reaction with *α*-diazooxime ethers **184** under copper catalysis conditions [90]. The reaction works well for a broad range of substituents and also allows for preparation of other *N*-heterocyclic compounds such as imidazopyridazines, imidazopyrimidines, and imidazopyrazines **187** (Scheme 44). A proposed reaction mechanism begins with conversion of *α*-diazooxime ether **184** to Cu-carbene **45A,** which is attacked by the pyridine nitrogen atom to form zwitterionic intermediate **45B**. Then, **45B** undergoes cyclization with regeneration of copper catalyst to give **45C,** which, after elimination of alcohol, affords imidazo[1,2-*a*]pyridine **185** (Scheme 45).

A rare example of another fused imidazole heterocycle prepared from the diazo substrate was presented in the work by Ueda and co-workers [91]. Theophyllines **191** were obtained along with cyanamides **192** from the diazobarbituric acid derivative **189** by treatment with amine **190** in the presence of Rh_2_(OAc)_4_. The imino diazo compound **189** was synthesized by oxidation of 7,8-diaminotheophyline **188** with Pb(OAc)_4_ (Scheme 46). Unfortunately, the method is characterized by low selectivity.

### 2.5. Pyrazoles (1H, 3H and 4H) and Indazoles

Pyrazole and indazole are common units in a number of synthetic products, such as tartrazine (food coloring agent), dipyrone (antipyretic and analgesic agent), sildenafil (drug used for the treatment of erectile disfunction), rimonabant (drug used to treat obesity), celecoxib (selective COX-2 inhibitors and non-steroidal anti-inflammatory drug, NSAID), fipronil (insecticide), pazopanib (tyrosine kinase inhibitor), and benzydamide (locally acting NSAID). Other biological activities of pyrazole- and indazole-containing compounds include antimicrobial, antihyperglycemic, pesticidal, leishmanicidal, antichagasic, anti-HIV, anti-depressant, and antiarrhythmic [92,93,94,95]. Among methods for the synthesis of pyrazoles diazo strategies are widely used. One commonly known approach is the 1,3-dipolar cycloaddition between diazo compounds and alkynes, or alkenes with the leaving group [96,97], including deacylative variant [14,15]. Another route is an intramolecular cyclization of vinyldiazo compounds [98]. Moreover, to date, a number of other methods involving diazo compounds have been developed.

#### 2.5.1. 1*H*-Pyrazoles

##### Synthesis of 1*H*-Pyrazoles by 1,5-Electrocyclization of Vinyldiazo Compounds

A mechanism of this transformation is depicted in Scheme 47. It involves 1,5-electrocyclization of the vinyldiazo compound **193** with subsequent tautomerization to yield 1*H*-pyrazole **195** [99].

In the works by the Nikolaev group, it was shown that configuration of double bond in vinyldiazo compounds significantly affects their reactivity in this reaction (Scheme 48) [100,101]. *trans*-Vinyl diazo compounds **196** readily underwent electrocyclization under moderate temperature (80 °C), while their *cis*-counterparts **198** produced pyrazoles **199** only at elevated temperatures (up to 120 °C) or decomposed with formation of furans **200**. Moreover, the authors demonstrated that fluoroalkyl-substituted vinyldiazo carbonyls **201** (both *trans* and *cis*) failed to give any pyrazole products, furnishing only carbene-derived products on heating. Interestingly, the diazo enol ***enol*-202** formed in situ can also undergo this transformation, affording 4-hydroxypyrazole **203**.

An attempt to prepare carbapenem precursor **206** by treatment of diazo carbonyls **204** with DBU and *β*-lactam **205** unexpectedly led to the formation of hydroxypyrazoles **207** (Scheme 49) [102]. From the mechanism standpoint, the 1,5-electrocyclization of enolate **50A** likely gave pyrazolium anion **50C,** which reacted with imine **50D,** which was, in turn, formed by DBU-promoted elimination of acetic acid from *β*-lactam **205** (Scheme 50). The direct regiospecific synthesis of *N*-arylpyrazoles by the co-catalyzed reaction of vinyl diazo carbonyls and aryl diazonium salts has also been reported [103].

The synthesis of mono-, di-, and tri-substituted pyrazoles **209** from vinyldiazo compounds **208** was recently reported by Dikermann and co-workers (Scheme 51) [104]. The reaction proceeds smoothly on heating in trifluoromethylbenzene affording desired products in yields of up to 94%. The synthesis of 5-arylsubstituted pyrazoles by the one-pot diazo transfer-cyclization approach has also been reported [105]. Separate examples of cyclization of vinyldiazo compounds to pyrazoles are presented in a number of works by other authors [106,107,108,109,110,111].

A cascade electrocyclization-Michael addition process for the synthesis of pyrano[3,2-*c*]pyrazol-7(1*H*)-ones **211** from 2-diazo-3,5-dioxo-6-ynoates (ynones) **210** was developed by Deng and co-workers [112]. The reaction tolerates a broad range of substituents giving generally excellent yields of annulated pyrazoles. Apparently, the reaction begins with the formation of enolate **52A,** which undergoes a 6-*π* electrocyclic ring closure to produce 3*H*-pyrazole intermediate **52B**. The following Michael addition of enolate oxygen to triple bond provides carbanion **52C**. Subsequent protonation and [1,5]-*H* shift affords the desired product **211** (Scheme 52).

Electrocyclization of vinyl diazo compounds **212** can trigger subsequent rearrangements, such as the Cope rearrangement and 1,3-sigmatropic shift as was discovered by Babinski and co-workers [113,114]. The initially formed 3*H*-pyrazole **53A** (although it can undergo the van Alphen-Hüttel rearrangement to become aromatic) in this case undergoes proton shift to give intermediate **53B**. The latter has two pathways to transform itself to aromatic pyrazole. In path I, a rare [3,3]-aromatic Cope rearrangement can occur, followed by re-aromatization to afford pyrazole **213**. In path II, intermediate **53B** can suffer a 1,3-alkyl shift to give pyrazole **214** (Scheme 53) [114]. Path II presents a radical reaction, so it can be facilitated by addition of radical initiator (such as AIBN), whereas path I is favored in presence of radical inhibitor (such as BHT). Although the scope of this tunable reaction is limited to benzyls with electron-withdrawing groups in positions 2 and 4, it gives access to two types of fully substituted pyrazoles **213** and **214**.

##### Miscellaneous Methods for Pyrazole Ring Construction from Vinyl Diazo Compounds

In 2013, the Doyle group devised a regiospecific one-pot, two-step method for the synthesis of pyrazoles **217** via the rhodium-catalyzed vinylogous addition of donor-acceptor hydrazones **216** to enol diazo acetates **215** followed by Lewis acid catalyzed cyclization (Scheme 54) [115]. Numerous substituents were tolerated, although in the case of substituents other than hydrogen in the vinylogous position (R^1^), the yields were lower.

An unprecedented chemo- and regioselective gold-catalyzed cross-coupling of vinyl diazo compounds **218**/**221** and aryl diazo acetates **219**/**222** was described by Xu and co-workers [116]. The ligand-controlled reaction gave pyrazoles **220** and **223** in a position-switchable mode. If [IPrAuNTf_2_] was used as the catalyst, the products were pyrazoles **220** with the *N*-substituent adjacent to the aryl (or ethyl) group, while with [(ArO)_3_PAuCl] as the catalyst, the regioselectivity was reversed and the products were pyrazoles **223**. For the first mode of the reaction, the presence of an *ortho*-substituent in the aryl moiety of aryl diazo acetates **219** was essential, whereas the second mode tolerated substituents in all positions of the phenyl ring of **222** (Scheme 55).

Another example of gold-catalyzed reaction of vinyl diazo compounds for the synthesis of pyrazole derivatives was described in 2019 by Raj and Liu [117]. The procedure involving [5+4]-annulation between 2-alkynyl-1-carbonylbenzenes **224** and vinyl diazo ketones **225** afforded 4,5-dihydro-benzo[*g*]indazoles **226** in generally good yields and with high diastereoselectivity. These initial products can be further converted to aromatic pyrazoles **227** and **228**/**228’**. The possible reaction mechanism likely involves the gold-catalyzed formation of benzopyrilium cation **56B** followed by its [5+4]-cycloaddition with vinyldiazo ketone **225** to yield intermediate **56C**. The latter can be hydrolyzed to give intermediate **56E,** which then undergoes a disrotatory 6π-electrocyclization with subsequent diastereoselective protonation leading to the observed pyrazoline **226** (Scheme 56).

##### Synthesis of 1*H*-Pyrazoles from Diazo Compounds and Alkynes

In this section, only selected examples are considered that have not been reviewed previously.

In the classic 1,3-dipolar cycloaddition of diazo compound **229** to alkyne **230**, 3*H*-pyrazole **231** is initially formed and, in some cases, can be isolated (Section 2.5.2). However, it usually undergoes a prototropic shift or the van Alphen–Hüttel [118,119] rearrangement to produce 1*H*-pyrazole **232** (Scheme 57) [120].

In 2015, Sultanova and co-authors reported an efficient synthesis of 7-oxo-4,5,6,7-tetrahydropyrazolo[1,5-*c*]pyrimidines **234** by reaction of 3-diazopyrrolidones **233** with dimethyl acetylenedicarboxylate (DMAD) [121]. The reaction is believed to proceed through the formation of a spirocyclic 3*H*-pyrazole intermediate **58A** via 1,3-dipolar cycloaddition with subsequent ring expansion via the van Alphen–Hüttel rearrangement (Scheme 58).

An interesting rearrangement of aryl propargyl *α*-aryl diazo acetates **235** to 1,5-dihydro-4*H*-pyrazol-4-ones **236** and **237** was reported by the Doyle group in 2016 [122]. The gold-catalyzed 1,3-acyloxy rearrangement of **235** likely gives allene species **59A,** which may exist in equilibrium with **235**. Intermediate **59A** undergoes rearrangement to produce dihydropyrazolone **236** directly in a concerted fashion or stepwise through intermediate **59D**. Under the reaction conditions, compound **236** is in equilibrium with **237**, via 1,3-acyl migration. At room temperature, the ratio of **236** to **237** is generally about 3:1. It is worth mentioning that these dihydropyrazolones can act as acyl transfer reagent, and when reacted with amine, afforded 4-hydroxypyrazoles **238** (Scheme 59).

A rare example of synthesis of 4*H*-pyrazoles was reported in 2018 by Zhang and co-workers (Scheme 60) [123]. An intramolecular 1,3-dipolar cycloaddition of alkyne-tethered *α*-cyano diazoamides **239** with subsequent 1,5-carbonyl migration provided spiro-4*H*-pyrazole-oxindoles **240** in good to excellent yields (Scheme 60, path a). The reaction worked well even with terminal alkyne. *α*-Methyl diazoamide **239a** also gave the corresponding 4*H*-pyrazole **240a**, albeit in diminished yield. In addition, *α*-sulfonyl diazo compounds **241** were employed in the reaction but in this case, the product was 1*H*-pyrazole **242** resulting from 1,5-sulfonyl migration to the *N*-atom in the 3*H*-pyrazole intermediate **60A** (Scheme 60, path b).

A continuous-flow synthesis of fluoroalkyl-substituted diazomethanes **244** from the corresponding amines **243** and their application in [3+2]-cycloaddition reactions with various alkynes and alkenes was described by Mertens and co-workers (Scheme 61) [124]. The method proved to be efficient for the preparation of fluoroalkyl-substituted 1*H*-pyrazoles **248**, pyrazolines **249**, and 3*H*-pyrazoles **250** with absolute regioselectivity in generally good to excellent yields. An approach for the preparation of mono-, bis-, and tris(trifluoromethyl)-substituted pyrazoles from ethyl 2-diazo-3,3,3-trifluoropropanoate has also been reported by the Reissig group [125].

#### 2.5.2. 3*H*-Pyrazoles

Relatively stable 3*H*-pyrazoles, containing two substituents in position 3, are commonly prepared from disubstituted diazo compounds. Although, they are still prone to rearrange to more stable products via, for instance, the van Alphen–Hüttel rearrangement or denitrogenative transformations [126].

In 2007, Strakova and co-workers reported the synthesis of spiro-3*H*-pyrazoles adducts **252** and **254** by [3+2] cycloaddition of DMAD to diazo compounds **251** and **253**, respectively (Scheme 62) [127]. The reaction proceeded smoothly at room temperature; however, the reaction took four days to go to completion.

A regio- and stereoselective synthesis of spiro-3*H*-pyrazole-penicillanates **257** was achieved by Santos and co-workers [128]. Benzhydryl esters provided better yields than their benzyl counterparts. The observed stereoselectivity likely resulted from the alkyne **256** approaching diazo *β*-lactam **255** from the less sterically hindered *α*-side (Scheme 63).

In 2015, Shelke and Suryavanshi described a method for the synthesis of 3,3′-spiro-phosphonylpyrazole-oxindoles **260** from methylidene indolinones **258** and the Bestmann-Ohira reagent (BOR) **259** (Scheme 64a) [129]. The regioselective reaction tolerated a broad range of substituents and afforded good yields of the desired products. Moreover, a one-pot procedure with generation of indolinones **258** in situ by the Wittig reaction of readily available isatins **261** and phosphonium ylides **262** was developed (Scheme 64b). The mechanism of the reaction likely involves the deacylation of BOR **259a** by the methoxide ion with subsequent 1,3-dipolar cycloaddition of the diazo-phosphonate anion (arising from **64A**) to methylidene indolinone **258a** to form intermediate **64B**. Subsequent protonation of **64B** and proton transfer produces intermediate **64D,** which is oxidized by air to afford the final compound **260a**.

In a study devoted to the exploration of fluorescence turn-on cycloadditions of dibenzocyclooctyne derivative **263** with various 1,3-dipoles, including di- and mono-substituted diazo compounds **264** and **266**, Friscourt and co-authors prepared 3*H*-pyrazole and 1*H*-pyrazole cycloadducts **265** and **267** [130]. The reaction is believed to be strain-promoted, which resulted in a short reaction time (2 h), even at room temperature. 3*H*-Pyrazole adducts **265** showed low quantum yields of the fluorescence whereas 1*H*-pyrazole adducts **267** exhibited a 160-fold fluorescence enhancement over the starting compound **263** (Scheme 65).

An unusual three-step approach to 5-substituted spiro-3*H*-pyrazoles was reported in 2008 by the Khidre group [131]. Diazo indan-1,3-dione **268** reacted with triphenyl(vinyl)phosphonium bromide **269** to give pyrazoline **270,** which was then converted to phosphonium ylide **271** upon treatment with EtONa. Reaction of **271** with benzaldehyde afforded spiro-3*H*-pyrazole derivative **272**. On the other hand, the involvement of triphenyl(prop-1-en-1-yl)phosphonium bromide (**273**) into the reaction allowed for the direct synthesis of 5-methyl-spiro-3*H*-pyrazole **274** (Scheme 66).

#### 2.5.3. Indazoles

A vast majority of methods for the preparation of indazoles from diazo compounds make use of a 1,3-dipolar cycloaddition reaction with an aryne species generated in situ. Among aryne precursors, *o*-silylaryl triflates are the most commonly used.

##### 1*H*-Indazoles

When a terminal diazo compound reacts with aryne, 3*H*-indazole is initially formed. Subsequently, it can undergo a prototropic isomerization to give 1*H*-indazole. In 2007, Jin and Yamamoto described a method for the synthesis of 1*H*-indazoles **277** and **278** by such an approach [132]. Using 1.2 equivalents of the diazo reagent **276** and KF/18-crown-6 in THF system provided *N*-unsubstituted indazoles **277** while using 0.5 equivalents of the diazo reagent **276** with CsF in acetonitrile led to *N*-arylated products **278**. As for asymmetrical silylaryl triflates, the complete regioselectivity was achieved only in the case of methoxy-substituted one (7-methoxy isomer **277a** was formed exclusively) whereas in other cases, a 1:1 mixture of regioisomers was obtained (Scheme 67).

When an internal diazo compound reacted with a benzyne intermediate, 3*H*-indazole intermediate **68A** is formed. For diazodicarbonyl compounds **279**, intermediate **68A** was prone to undergo acyl migration to give 1*H*-indazoles **281** as was shown by the Larock group [133]. On the other hand, benzyl-substituted diazo ester **282** and diazo diphenylmethane **284** gave stable 3*H*-indazoles **283** and **285** exclusively (Scheme 68).

An efficient protocol for the synthesis of *N*-substituted indazoles from the Bestmann–Ohira reagent (BOR) analogs **286**, silylaryl triflates **287,** and Michael acceptors **288** was developed by Phatake and co-workers [134]. A role of fluoride anion in this reaction was not only in the generation of the aryne intermediate, but also in the dephosphonylation of BOR. The cascade [3+2] cycloaddition of the acyl diazomethane anion to aryne following the aza-Michael addition process furnished 1,3-substituted indazoles **289** in generally high yields under very mild reaction conditions (Scheme 69).

In 2016, Ikawa and co-workers demonstrated that 2,4-bis(trimethylsilyl)-1,3-bis(trifluoromethanesulfonyloxy)benzene **290** could act as a 1,4-benzdiyne equivalent [135]. Moreover, aryne “triple bonds” could be generated in a sequential manner. Using this methodology, an exotic indazole derivative **293** was prepared in a two-step procedure with good overall yield (Scheme 70).

A tunable approach for the synthesis of 1*H*- and 3*H*-indazoles was reported by Chen and co-authors [136]. The product distribution depended on the structure of the phosphoryl group in *α*-substituted-*α*-diazophosphonates **294**. Bulky diisopropyl phosponates provided stable 3*H*-indazoles **296,** whereas less sterically hindered dimethyl phosphonates gave 1*H*-indazoles **297** (Scheme 71).

The ‘hexadehydro Diels-Alder’ (HDDA) reaction was employed in the synthesis of indazoles. In 2017, the Hoye group—in their work devoted to investigation of photochemical variation of this reaction—demonstrated that a highly functionalized 1*H-* and 3*H*-indazoles **300** and **301** can be constructed from a tetraalkyne derivative **298** and corresponding diazo compounds (Scheme 72a) [137]. While developing this methodology further, this group showed that an efficient domino HDDA reaction of polyalkynes (such as **302**) could be used for the synthesis of diverse polyaromatic systems, including indazole **303** (Scheme 72b) [138].

Other aryne precursors, including heteroaromatic ones, have also been reported to-date [139,140,141,142,143,144,145]. The aryne cycloaddition approach also found its application in the synthesis of biologically active indazoles [146].

As for other methodologies not involving arynes, Muruganantham and Namboothiri reported a reaction of BOR **259a** with nitroalkenes, including nitroethylene moiety as a part of aromatic or heteroaromatic ring (such as **304**) [147]. The process consisting of [3+2] cycloaddition followed by HNO_2_ elimination provides benzo- or pyrido-fused phosphonylindazoles **305** in moderate yields (Scheme 73). Unfortunately, the reaction failed to give any indazole products with simple nitrobenzene derivatives.

A 1,3-dipolar cycloaddition reaction of naphthoquinones **306** and diazo compounds **307** was used for the synthesis of naphthohydroquinone-derived indazoles **308** in a work by Tandon and co-workers [148]. Biological evaluation of obtained compounds showed that **308a** exhibited antifungal and antibacterial activities. The drawback of this approach is that indazoles **308** were obtained as a mixture with (1,4)-naphthoquinono-[3,2-*c*]-1*H*-pyrazoles **309** (Scheme 74).

##### 3*H*-Indazoles and 2*H*-Indazoles

When a 5-,6- or 7-membered cyclic diazocarbonyl compound **310** reacted with an aryne intermediate **75A**, the migration of a carbonyl moiety of initially formed spiro-3*H*-indazole **311** is restricted to 1,2-rearrangement since 1,3-migration is impossible in this case. Such a rearrangement typically gives 2*H*-indazole **312** (Scheme 75) [149]. Although these spiro-3*H*-indazoles **311** can be quite stable, some methods have been reported for their conversion to 2*H*-indazoles **312**.

In 2017, the Zhai group devised a protocol for the preparation of stable spiro-3*H*-indazoles by the cycloaddition of 3-diazoindolin-2-ones **313** to arynes generated from silylaryl triflates **314** [150]. The reaction tolerated electron-withdrawing and electron-donating groups in both indolinone and silylaryl triflate moieties providing spiro[indazole-3,3′-indolin]-2′-ones **315** in good to excellent yields. Under thermal conditions, these products underwent rearrangement to produce fused 2*H*-indazoles **316** (Scheme 76). The syntheses of various other spiro-3*H*-indazoles from 3-diazoindolin-2-one derivatives have also been reported in other works [151,152].

This methodology was further extended to carbocyclic diazo compounds **317** [153]. In this case, the rearrangement into 2*H*-indazoles **319** could be carried out not only under thermal but also under acidic conditions, even with weakly acidic SiO_2_ (Scheme 77). The reaction worked well with 6-membered cyclic diazo compounds, but in the case of 7-membered ones, pure 3*H*-indazole could not be obtained. Fortunately, performing the two-step reaction in a one-pot fashion allowed to prepare 2*H*-indazole fused with the seven-membered ring in a moderate yield. Noteworthily, the five-membered cyclic diazo compound failed to give any indazole products.

In 2012, Wang and Liu found that cationic Ag(I) species can dramatically affect the chemoselectivity of the reaction of terminal diazo carbonyls **320** with an excess of arynes [154]. Thus, in the presence of AgOTf, 2-aryl-2*H*-indazoles **322**. This result is in contrast to 1-aryl-1*H*-indazoles **278** which were obtained by Jin and Yamamoto in absence of any catalyst (*vide supra*, Scheme 67). The authors postulated the following mechanism for this reaction. Intermediate **78A** generated by initial cycloaddition is deprotonated by fluoride to give a carbonyl-stabilized anion **78B,** which is, in turn, trapped by Ag(I) cation to form (1*H*-indazol-1-yl)silver **78C.** The reaction of intermediate **78C** with benzyne is likely to produce the distinct arylation product **78D**. Subsequent elimination of silver with tautomerization furnishes the final 2*H*-indazole product **322a** (Scheme 78).

Not only «aryne» approaches have been employed for the synthesis of 2*H*-indazoles. In a study by the Lee group, 2*H*-indazoles were prepared from 2-iodoazoarenes/2-iodoaryltriazenes **323** and *α*-acetyldiazoacetate **151** by a one-pot tandem palladium-catalyzed deacylative cross-coupling/denitrogenative cyclization process [155]. The method was applicable to a broad range of substituents in the aryl moiety of **323** and afforded 2-substituted 2*H*-indazoles **324** in generally good yields (Scheme 79). Moreover, this protocol provided entry into indazoles with different substituents in the two aryl rings, which is not achievable when using «aryne» methodologies.

Azoxy compounds can be used for the synthesis of 2*H*-indazoles, as was shown by the You group [156]. The Rh(III)-catalyzed C-H-alkylation/intramolecular decarboxylative cyclization reaction of diazo esters **325** and aryldiazene oxides **326** provided 3-acyl-2*H*-indazoles **327** in a highly regioselective manner (Scheme 80). As in the previous case, both symmetrical and non-symmetrical diaryldiazene oxides were found suitable for the reaction. In addition, the protocol is also compatible with 2-alkyl-1-aryldiazene oxides giving *N*-alkyl-2*H*-indazoles in acceptable yields.

## 3. Azoles with Three Heteroatoms

### 3.1. 1,2,3-Thiadiazoles

A general approach to the synthesis of 1,2,3-thiadiazoles from diazo compounds is based on the reaction of diazocarbonyl compound **328** with a thionation reagent. The initially formed diazothiocarbonyl compound **329** is usually unstable and undergoes 1,5-electrocyclization to form 1,2,3-thiadiazole **330** (Scheme 81). This topic has been recently reviewed by Shafran and co-authors [16], therefore we chose not cover it in the present review.

### 3.2. 1,2,3-Triazoles

Since the revolutionary discovery of Copper-Azide-Alkyne Cycloaddition (CuAAC) by Meldal [157] and Fokin and Sharpless [158], this reaction has become the most widely used method for the preparation of 1,2,3-triazoles. Due to the great versatility and simplicity of this approach, it found application in many fields of chemical science, such as bioorthogonal reactions [159], design of functional coatings [160], preparation of organic dyes and fluorophores [161], etc. It has also become a handy tool in medicinal chemistry and drug design for the preparation of triazole-containing bioactive molecules possessing anticancer, antimicrobial, anti-tubercular, antiviral, antidiabetic, antimalarial, anti-leishmanial, and neuroprotective activities [162].

Standard CuAAC is suitable for terminal alkynes and allows the synthesis of 1,4-disubstituted 1,2,3-triazoles, but the regioselectivity can be reversed by using a ruthenium catalyst for synthesis of 1,5-disubstituted triazoles. Additionally, RuAAC also works well with internal alkynes providing fully substituted triazoles, although in this case, regioselectivity cannot always be reliably controlled [163]. In contrast with this method, a number of azide-free methodologies for regiospecific synthesis of variously substituted 1,2,3-triazoles have been developed [164,165], including the «diazo» approach, specifically, the Wolff cyclocondensation. Moreover, for the latter reaction, common Lewis acids are used and often no transition metal catalyst is needed.

#### 3.2.1. 1,2,3-Triazoles via Cyclization of Imino Diazo Compounds

The Wolff cyclocondensation is a reaction of diazocarbonyl compound with amine leading to 1,2,3-triazoles [166]. The mechanism of this reaction is depicted in Scheme 82. The condensation of amine **332** with diazo compound **331** carbonyl group, usually occurring under acidic catalysis, gives diazo imine intermediate **333,** which is in equilibrium with 1,2,3-triazole **334**. When the R group in the imine moiety is not electron-withdrawing, this equilibrium is completely shifted towards triazole. The disadvantages of this approach in comparison with CuAAC include its questionable bioorthogonality and often harsh reaction conditions.

In a study devoted to the synthesis of potential anti-tubercular agents, Costa and co-workers employed a reaction of diazo malonaldehyde **335** with aniline hydrochlorides **336** conducted in water at room temperature to afford 1,2,3-triazoles with a 1,4-substitution pattern **337** (Scheme 83) [167]. It should be noted that amine hydrochlorides themselves provide an acidic medium, which facilitates the generation of intermediate imine species. Such 1-aryl-1,2,3-triazole-4-carbaldehydes **337** can be further converted into difluoromethyl-substituted triazoles **338** in excellent yields simply on treatment with diethylaminosulfur trifluoride (DAST).

In contrast to diazo aldehydes, the reaction of amines with *α*-diazoketones usually requires the use of elevated temperature and the addition of Lewis acids in stoichiometric amounts [168]. *α*-Diazo-*β*-oxosulfones **339** can also be involved in this reaction, as was shown by the Krasavin group [169]. Heating at 80 °C in chlorobenzene in the presence of 1.5 equivalents of TiCl_4_ turned out to be the best conditions for this transformation, although the desired 4-sulfonyl-1,2,3-triazoles **341** were obtained in moderate yields (Scheme 84).

An interesting effect of intramolecular hydrogen bonding activation of carbonyl group in *α*-diazo-*β*-oxoamides **342** was described in 2012 by Wang and co-workers (Scheme 85) [170]. This effect allowed the use of catalytic amounts (20 mol.%) of Lewis acid, affording diversely substituted triazole-4-carboxamides **344** in good yields (Scheme 85a). The method works well with both aliphatic and aromatic amines, as well as diamines such as **345** (Scheme 85b), substituted hydrazine, hydroxylamine, and even ammonium acetate (Scheme 85c). In the latter case, 1*H*-triazole-5-carboxamides **347** were obtained. This methodology was later used for the preparation of some triazole-containing polymers [171].

Even poorly nucleophilic amines, such as aminofuroxans **348**, can be employed for this reaction, as was shown by Fershtat and co-authors [172]. The use of 20 mol.% of boron trifluoride etherate at room temperature proved to be the optimal conditions for this substrate, although the desired triazoles **350** were obtained in low to moderate yields. It is worth mentioning that in this case, *α*-diazodiketones gave better yields than *α*-diazo-*β*-ketoesters (Scheme 86).

An intramolecular variant of this reaction usually requires milder conditions and often no use of a Lewis acid catalyst. In 2019, Santiago and Burtoloso reported the synthesis of fused bicyclic 1,2,3-triazoles **355** from *γ-N*-protected amino diazo ketones **354** [173]. The starting diazo compounds were prepared in four steps from γ-nitro esters **351**. The cyclization step was performed by gently refluxing the diazo ketones **354** in a methanolic aqueous solution of potassium carbonate. The yields on the final step were good to excellent, the only diminished yield was observed with *α*-methyl-*α*-diazoketone **354a** (Scheme 87).

Combining the halo, keto, and diazo functions in *α*-diazo-*β*-keto-*γ*-haloesters **356** has a potential for constructing fused triazole systems. This potential was realized in a recent work by Krasavin and co-workers who reacted these diazo compounds with aminothiols **357** (Scheme 88) [174]. Initially, halogen is substituted by a more nucleophilic thiol group so that the subsequent imine formation becomes an intramolecular reaction and requires rather mild conditions to proceed. Both aliphatic and aromatic aminothiols worked well, providing 6,7-dihydrotriazolothiazines **358** in good to excellent yields. Attempts to perform this reaction with thiosemicarbazides **359** only resulted in the formation of substitution product, which, however, underwent rapid cyclization into triazole **360** in pure acetic acid (Scheme 88).

The reaction of diazo ketone **151** with substituted hydrazines **361** occurs under mild conditions and acetic acid as acidic promoter is usually sufficient for formation of hydrazone intermediates **89A** [175]. Jordão and co-workers used this approach in their work on the synthesis and antiviral testing of *N*-amino-1,2,3-triazoles **362** and their hydrazides **363** against the Cantagalo virus (Scheme 89) [176]. The best yield was achieved with unsubstituted phenylhydrazine. The yield decreased as electron-withdrawing substituents were introduced. These triazoles were further modified in order to obtain potential anti-HSV-1 [177] and anti-candidiasis [178] agents.

The steps sequence in a «diazo» approach to 1,2,3-triazoles can be reversed so that imine is formed first and then undergoes a diazo transfer reaction. In 2016, the Emmanuvel group developed a one-pot procedure for the synthesis of *N*-aminotriazoles **365** from *β*-ketocarbonyl compounds **364** and *tert*-butyl carbazate [179]. The reaction also worked well with 2,4-dinitrophenylhydrazine **366**. However, when phenylhydrazine was involved into this process, pyrazolone **368** formation was observed instead of the desired 1,2,3-triazole (Scheme 90). In other work by this group, authors were able to isolate the intermediate diazo hydrazones using different diazo transfer conditions [180]. A deacylative variation of diazo transfer on enaminones for the synthesis of 1,5-disubstituted triazoles have also been reported to date [181].

5-Aminotriazoles **370** can be obtained by performing diazo transfer on amidine hydrochlorides **369** as was demonstrated by Dankova and co-authors [182]. The reaction proceeded regioselectively with asymmetric amidines, except for **369b** when a 3:1 mixture of isomeric triazoles **370b** and **370b’** was obtained (Scheme 91).

*α*-Diazoamides are known to undergo cyclization under basic conditions to give salts of 5-hydroxy-1,2,3-triazoles [183]. In 1984, Murrey-Rust and coworkers developed a simple method for the preparation of 5-hydroxytriazole ammonium salts **373** by treatment of dimethyl diazomalonate **371** with an excess nuber of amines **372**—neat or in toluene solution. (Scheme 92) [184]. The role of amine in this reaction is both as a nucleophile for the formation of diazo amide and as a base for the subsequent cyclization. The limitation of this approach is that aromatic amines fail to react, which is consistent with reduced nucleophilicity (and basicity) of the amino group in aromatic amines. This methodology has been used recently for the preparation of 5-hydroxytriazole derivatives acting as a potential carboxylic acid bioisosters [185,186]. Hydroxytriazole salts can be acidified to afford free 5-hydroxytriazoles **374**, but these compounds are often unstable and undergo ring opening to form parent diazo amide species [187]. Some mechanistic investigations of these equilibrium processes have been reported elsewhere [188,189].

The information on synthesis of *N*-hydroxytriazoles from diazo compounds is scarce. In a report by Jiang and co-workers devoted to the synthesis of some *N*-hydroxytriazoles and their evaluation as peptide coupling reagents, 1-hydroxy-5-methyltriazole-4-carboxylate **375** was prepared by the Wolff condensation of *α*-diazo-*β*-ketobutanoate **151** with hydroxylamine [190]. This procedure, however, did not work well for the synthesis of 5-unsubstituted 1-hydroxytriazole **379**, so an alternative route had to be devised. Ethyl diazoacetate **82** was coupled with Vilsmeier reagent **376** and the product **377** was treated with hydroxylamine. The cyclization of diazo oxime **378** thus obtained was carried out either by refluxing it in benzene or by storing its solution in chloroform at room temperature for 2 days. The overall yield of the target *N*-hydroxytriazole **379** was fair (Scheme 93).

Several approaches to the synthesis of intermediate diazo imines have been reported to date. For example, Regitz and Schoder reported the synthesis of triazoles **382** in which diazo imine intermediate **94A** was prepared by iminomethylation of terminal diazo compounds **380** with formamidine **381** under reflux in acetonitrile (Scheme 94) [191].

5-Halo-substituted triazoles **384** can be prepared from *α*-diazo-*β*-ketonitriles **383** by treatment of the latter with HHal, as was described in a work by the Mokrushin group [192]. The reaction is believed to proceed via diazoimidoyl chloride intermediate **95A**. The presence of an acyl group at the *α*-position of diazo nitrile proved to be crucial for this reaction (Scheme 95). *α*-Phosphonyl *α*-diazoacetonitriles was also shown to undergo this transformation.

#### 3.2.2. 1,2,3-Triazoles via Formal Cycloaddition to Multiple CN bonds

This section describes methods that do not rely on the formation of diazo imine intermediates. In 2010, Chen and co-workers reported a method for the construction of fully substituted triazoles **387** by a DBU-mediated reaction of diazo compounds **385** with *N*-PMP aryl imines **386** (Scheme 96) [193]. The best yields were obtained with EWG-substituted aryl imines, diminishing in the case of halogen or EDG-substituted ones.

This reaction can be performed in a three-component fashion using aldehyde **388**, amine **389**, and BOR (**390**) as the diazo component as was shown by the Mohanan group [194]. The method allows the diastereoselective synthesis of either triazolines **391** or triazoles **392**, depending on the nature of the amine and the aldehyde used. The tentative mechanism is as follows. BOR (**390**), upon deacylation, reacted with imine **97B** to give the intermediate **97C**. Subsequent *5-endo-dig* cyclization and proton transfer furnished the triazoline **391**. When the aldehyde and the amine are both aromatic, **391** can undergo a spontaneous air oxidation to form triazole **392** (Scheme 97). Interestingly, in contrast with the previous example, aldehydes bearing EWG-substituents in the aryl moiety failed to deliver the product. An analogous protocol for the preparation of triazole-4-sulfones have been reported by this group [195].

Li and co-workers used CuBr_2_ oxidation of secondary amines, namely *N*-arylglycine derivatives **393**, for the in situ generation of imines [196]. Their reaction with diazo compounds **394** afforded fully decorated triazoles **395** in moderate to high yields. Mechanistically, the reaction proceeds through iminium salt **98A,** which upon cycloaddition with diazo compound **394** forms intermediate **98B**. Subsequent oxidation, probably by copper ions, produces the triazole product **395** (Scheme 98).

Silver-catalyzed [3+2] cycloaddition of diazo compounds **396** to isocyanides **397** was reported by Wang and co-workers [197]. The reaction proceeded smoothly to give 1,4-disubstituted triazoles **398** with complete regioselectivity (Scheme 99). This protocol is compatible with both EWG- and EDG-substituted aryl isocyanides. However, when aliphatic isocyanides were used, the yields were decreased. As for the diazo compounds, trifluorodiazoethane, diazo esters, and trimethylsilyl diazomethane can be used, although the latter provides the desired triazoles only in moderate yields.

### 3.3. 1,2,4-Triazoles

1,2,4-Triazoles possess the biological profile that is somewhat similar to that displayed by 1,2,3-triazoles. Moreover, medicinal agents containing 1,2,4-triazole moiety exhibit numerous biological activities such as antifungal, antiparasitic, anxiolytic, antidepressant, anticonvulsant, etc. [198]. Besides, 1,2,4-triazole derivatives are used as ligand in coordination chemistry [199] and as a building blocks for the construction of coordination polymers and metal organic frameworks [200]. Classical methods for the synthesis of 1,2,4-triazoles include condensation of substituted hydrazines with imides, reaction of amides with hydrazides, and coupling of amidrazones with acyl chlorides, although many other methodologies have been developed [201]. However, there are only few reports on the synthesis of 1,2,4-triazoles using diazo compounds.

A protocol for the synthesis of 1,2,4-triazole-3-carboxylates **401** from imines **399** and diazo ketones **400** was devised by Zhao and co-wrokers [202]. The reaction involved a nucleophilic attack of anion **100A**, generated upon deprotonation of imine **399** with DBU, onto a terminal diazo nitrogen of **400** followed by proton transfer and tautomerization to give the hydrazone intermediate **100D**. Subsequent cyclization and air oxidation furnishes the final triazole product **401** (Scheme 100). Interestingly, only terminal *α*-diazoketones produced 1,2,4-triazoles, whereas diazo esters gave tetrahydro-1,2,4-triazines.

In 2018, the Chen group described a base-mediated reaction of diazo oxindoles **402** with oxazol-5-(4*H*)-ones **403** affording 3-(1*H*-1,2,4-triazol-1-yl)indolin-2-ones **404** in reasonable yields (Scheme 101) [203]. The best results were obtained with electron-donating R^1^ groups on the phenyl ring of **402**. As for the oxazolones **403**, electron-poor aryl rings as the R^3^ substituents provided the desired product in higher yields in comparison with electron-rich ones. Noteworthily, when 4-unsubstituted oxazolone **403** (R^4^ = H) was used, the reaction failed to give any triazole product. Attempts to perform the reaction with diazo dicarbonyl compounds and a vinyl diazo compound did not work as well. The mechanism of the reaction is presumably similar to the previous example.

Radical reactions of diazo compounds have also been employed for the assembly of hydrazones with subsequent cyclization to 1,2,4-triazoles. Chan and co-workers developed a photoredox decarboxylative C(sp^3^)−N coupling of *N*-hydroxyphthalimide (NHPI) esters with *α*-diazoacetates **406** for the synthesis of hydrazones [204]. When *N*-acetyl amino acids derived NHPI esters **405** were used, the initially formed hydrazones **102A** underwent cyclization to give 1,2,4-triazoles **407** in good yields. Mechanistically, the reaction begins with photoexcitation of Rose bengal (RB) with 585 nm yellow LED irradiation. An excited state RB* reacts with Hantzsch ester **408** by single-electron transfer (SET) process to generate the Rose bengal radical anion (RB^•−^), which reduces the NHPI ester **405** to form radical anion **102B**. Subsequent C–C bond fragmentation gives alkyl radical R^•^, which reacts with the terminal diazo nitrogen of **406** to produce intermediate **102C**. Following hydrogen atom transfer (HAT) from the cationic radical, Hantzsch ester **102D** provides hydrazone **102A**. Finally, **102A** undergoes cyclization to afford the desired 1,2,4-triazole product **407** (Scheme 102).

A three-component protocol for the preparation of 1,2,4-triazoles **413** from aryl diazonium salts **410**, diazo esters **411**, and nitriles **412** was developed by the Wan group [205]. EDG-substituted aryl diazonium salts performed better than EWG-substituted ones. Based on preliminary mechanistic studies and DFT-calculations, the authors suggest the following mechanism for the reaction. Nitrile **412** most likely attacks copper carbene **103A** to form intermediate **103B,** which then undergoes a highly regioselective [3+2] cycloaddition to diazonium salt **410** to afford cyclic intermediate **103C**. The latter suffers base-mediated isomerization to deliver the desired 1,2,4-triazole **413** (Scheme 103).

Diazo azoles, namely diazo pyrazoles **414** and diazo imidazoles **417**, were employed for the synthesis of *N*-azolyl-1,2,4-triazoles **416** and **418** in a study reported by Sadchikova and co-workers [206]. The reaction of **414** and **417** with ethyl isocyanoacetate **415** proceeded smoothly to afford the desired products in good to excellent yields (Scheme 104). Interestingly, when a primary or secondary amide moiety was present in the diazo heterocyclic substrate, the reaction gave azolotriazinones instead of triazoles.

## 4. Azoles with Four Heteroatoms

### Tetrazoles

Tetrazole derivatives are known to be high-energy compounds. They have found application in explosives, propellants, and pyrotechnics. In these fields, tetrazoles are used either as individual compounds [207] or as complexes with metals, which are called energetic metal-organic frameworks (EMOFs) [208]. Several tetrazoles have been prepared which outperform classic high performance explosives, such as RDX [207]. As for medicinal chemistry, 5-substituted tetrazoles are commonly considered as carboxylic acids bioisosteres whereas 1,5-disubstituted ones are used as amide surrogates. Moreover, tetrazoles possess remarkable metabolic stability. According to the Drug Bank, there are more than 40 drugs that exhibit antimicrobial, anticancer [209], antiallergic, hypertensive, nootropic, and other biological activities [210]. The general approach for the synthesis of 1-unsubstituted and 1,5-disubstituted tetrazoles is the reaction of nitriles or imidoyl derivatives with inorganic or organic azides, respectively [210,211]. In contrast to these methodologies, the synthesis of tetrazoles using diazo compounds provides access to an alternative regioisomer, namely, 2,5-disubstituted tetrazole, with high regioselectivity.

In 1982, Aoyama and Shioiri reported a reaction of a 2-fold excess of lithium trimethylsilyl diazomethane **419** with methyl esters **420,** which afforded 2-substituted-5-TMS-tetrazoles **421** in generally good yields [212]. The authors propose the following mechanism for the reaction. Lithium derivative **419** initially attacks the ester carbonyl carbon of **420** producing intermediate **105A**. Either the 1,3-dipolar cycloaddition of **105A** to **419** or the nucleophilic attack at the terminal diazo nitrogen of **105A** with a second molecule of **419,** followed by cyclization, leads to the formation of intermediate **105B**. The latter is hydrolyzed during the aqueous work-up to give the desired tetrazole products **421**. Similar «dimerization» of *α*-diazophosphonates has also been reported (Scheme 105) [213].

In recent years, an alternative «diazo» approach for the preparation of tetrazoles, which is based on the reaction of terminal diazo compounds with aryl diazonium salts, has been extensively studied. In 2015, the Ma group for the first time described a silver acetate-catalyzed [3+2]-cycloaddition of diversely substituted aryl diazonium tetrafluoroborates **422** and 2,2,2-trifluorodiazoethane **423** for the synthesis of 2-aryl-5-trifluoromethyltetrazoles **424** [214]. The reaction tolerated a wide range of electron-donating and electron-withdrawing substituents in the aryl ring. Some heterocyclic diazonium salts were also found to be compatible with this protocol. Moreover, the reaction could be performed in a one-pot diazotization/cycloaddition sequence starting from anilines **425**. In this case, electron-deficient substrates showed better yields in comparison with electron-rich ones (Scheme 106).

Further developing the above methodology, the same group demonstrated its utility for the synthesis of CF_2_-functionalized tetrazoles. ((2-Diazo-1,1-difluoroethyl)sulfonyl)benzene **426** was used as a diazo reagent to afford the desired tetrazoles **427** in good to excellent yields [215]. As in the previous example, the reaction can also be performed in a one-pot fashion. More importantly, the obtained tetrazoles **427** can be easily converted to the medicinally relevant CHF_2_-substituted tetrazoles **428** either by treatment with *t*BuOK or by reduction with LiAlH_4_ (Scheme 107).

In a 2016 study by Patouret and Kamenecka, trimethylsilyl diazomethane **42** was involved into this reaction in the presence of CsF to obtain 5-unsubstituted 2-aryltetrazoles **429** [216]. In this case, CF_3_CO_2_Ag was found to be the silver source of choice. The use of a stoichiometric amount of silver salt (1.2 eq.) and low temperature (−78 °C) proved to be crucial for a successful reaction. When the reaction was carried out in the absence of CsF, 5-TMS-substituted tetrazole **430** could be isolated. The latter can be used for the preparation of 5-bromotetrazole **431,** which has the potential for further modifications via cross-coupling reactions (Scheme 108).

*α*-Diazoketones along with *α*-diazoamides (**433**) were first employed for this reaction by Krasavin and co-workers to obtain 2-aryl-5-acyltetrazoles **434** in moderate to good yields (Scheme 109) [217]. In this case, silver nitrate as a catalyst showed the best results. As a diazonium reagent, aryl diazonium tosylates **432** were used as they are more stable and convenient to handle compared to the respective tetrafluoroborates. The reaction was found to be not particularly sensitive to substituent effects in the diazonium moiety. As for the diazo portion, aromatic and heteroaromatic diazo ketones performed better in comparison with aliphatic diazo ketone and diazo amides. Recently, *α*-diazoesters [218] and the Seyferth–Gilbert reagent [219] have been involved in this reaction by the Ma group.

In a 2020 work by the Krasavin group, it was demonstrated that (hetero)aryl-, alkylsulfonyl, and aminosulfonyl diazomethanes **435** are also applicable for this reaction [220]. Thus, the rare 2-aryltetrazol-5-yl sulfones and previously not described in the literature 2-aryltetrazol-5-yl sulfonamides **436** were prepared in generally good yields. However, diminished yields were observed with aryl diazonium tosylates **432** bearing electron-donating or *ortho*-substituents. The following plausible mechanism was suggested (which can be presumably extended also to the examples discussed above). Diazo substrate **435** was deprotonated by the base and bound by the Ag cation to give intermediate **110A**. The latter coordinated with diazonium salt in a way that unambiguously determines the regiochemistry of the final product, i.e., providing intermediate **110C**. Subsequent silver-assisted cycloaddition with the liberation of the catalyst delivers the desired tetrazole **436** (Scheme 110).

## 5. Conclusions

As we have demonstrated in this review, diazocarbonyl and related compounds provide rapid and efficient access to a wide range of various azoles. Among the most thoroughly studied are approaches to 1,3-oxazoles, pyrazoles, and 1,2,3-triazoles. Some of these have even become common synthetic methodologies in the organic chemist’s toolbox. On the other hand, the synthesis of 1,3-thiazoles, imidazoles, 1,2,3-thiadiazoles, and 1,2,4-triazoles is less developed. To date, information on the synthesis of isoxazoles from diazocarbonyl compounds has been limited to only a few reports. Most recently, great progress has been achieved in the synthesis of tetrazoles from various diazo compounds. However, it is obvious that the synthetic potential of diazo compounds in this area is far from its full disclosure. In addition, despite the extensive research, some mechanistic details are still not clear and need further investigation. The main advantage of «diazo» approaches to azoles is their high, often absolute regioselectivity. The utility of these methods is emphasized by numerous examples of the use of diazo compounds at key steps of the total synthesis of azole-containing natural products. Given the easy availability of diazo substrates, the described methodologies represent a valuable alternative to more conventional methods.

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
