# Peer review of "Diazocarbonyl and Related Compounds in the Synthesis of Azoles"

_molecules, 2021, doi:10.3390/molecules26092530_

Round 1
Reviewer 1 Report
The review effectively describes Diazocarbonyl derivatives and related compounds in the synthesis of Azoles. The work is well done and deserves to be published in the journal Molecules
Author Response
Thank you. We greatly appreciate your supportive review. The English has been polished throughout the manuscript.
Reviewer 2 Report
Krasavin and co-workers described a well-detailed review of diazocarbonyl and related compounds in the synthesis of azoles. Azoles are an important class of heterocyclic five-membered ring compounds and their synthesis represent an interesting and challenging endeavor in the synthetic community. This is a well-written review with 110 reaction schemes and summarizes numerous literature works (220 references) on the synthesis of all types of azoles from diazocarbonyl, oxazoles, thiazoles, imidazoles, pyrazoles, triazoles and tetrazoles.
Minor suggestion: The abstract can be improved to reflect the extensive work in this review. The authors can add a couple more of sentences to highlight the main areas of the review which will give readers a quick overview and what to expect for this review.
Recent work: This is FYI for the authors for a recent paper related to their review which they might consider to include.
- β‐Diazocarbonyl Compounds: Synthesis and their Rh(II)‐Catalyzed 1,3 C−H Insertions; 2021; https://doi.org/10.1002/anie.202015077
Typo: Scheme 36, b) Bis-imidazoles
Author Response
- Abstract has been improved.
- Regarding biimidazoles (Scheme 36): in the original paper these compounds were called "biimidazoles" (similar to bipyridines), so we used this name to describe them. https://doi.org/10.1002/adsc.201600574
Lines 471, 476, 477, 489, 492, 495.
Reviewer 3 Report
The various preparative methods of various azoles from diazocarbonyl and related compounds have been reviewed comprehensively in this review. The review is well organized and written with excellent structural and scheme drawings. It is very useful for organic chemists work in this field. It can be accepted for publication after a minor revision.
Additional comments
1. dinitrogen -- nitrogen
2. tBu t should be in italic in all schemes.
3. A, B, C, and D were used as intermediates numbers in different schemes. It is not good style. One letter should be used for only the same one structure. It is possible good choice to add scheme number before the letters A, B, C, and D.
4. Some structures without compound numbers, such as those in Scheme 82. Please check all schemes.
5. Generally, the same structure should be labeled with the same number in whole manuscript. Please check the whole manuscript and unify the numbers. In this review, some compounds have different numbers in different schemes, they have new numbers when they appear in different schemes.
Author Response
- Dinitrogen – nitrogen correction has been made. Line 17.
- tBu t has been checked and corrected.
- All numbers (including numbers of intermediates) in all schemes have been corrected.